# Cross-Anatomy Inference of Deep Learning-Based MRI Acceleration Methods

**Aleksandr Belov**[1,2]                                          ALEKSANDR.BELOV@PHILIPS.COM

**Joël Valentin Stadelmann**[1]                                  JOEL.STADELMANN@PHILIPS.COM

**Dmitry V. Dylov**[2]                                            D.DYLOV@SKOLTECH.RU

[1] *Philips Innovation Labs RUS, Moscow, Russian Federation*

[2] *Skolkovo Institute of Science and Technology, Moscow, Russian Federation*

## Abstract

MRI scanners capture images of excellent soft-tissue contrast but involve long acquisition times, which can be mitigated by acquiring undersampled or low-resolution k-space data. Although image-to-image translation and superresolution techniques can correct the artifacts caused by the incomplete k-space, they are known to be anatomy-specific, requiring validation and train sets to belong to the same body part. This short paper covers the causes of the lowered reconstruction quality in the cross-anatomy inference task and provides suggestions for their compensation.

**Keywords:** Fast MRI, Superresolution, Image-to-image translation.

## 1. Introduction

Magnetic Resonance Imaging (MRI) is a modality of choice for a wide range of clinical cases as it is non-invasive and does not expose the patient to harmful radiation. The acquisition of the raw ($k$-space) MRI data typically lasts 15-60 minutes, during which the patient must remain motionless. If the patient cannot remain still, artefacts appear on the image, often demanding a complete re-scan. Furthermore, the long acquisition times limit the applicability of MRI to dynamic imaging, such as that of the abdomen or the heart.

The scan duration can be reduced by incomplete $k$-space sampling, such as in compressed sensing (Debatin and McKinnon, 1998) or in adaptive intelligence (Pezzotti et al., 2020). However, deep learning (DL) based method usually restrain their evaluation to the anatomical site on which they were trained (Darestani et al., 2021; Wang et al., 2021). In this short paper, we present DL models that combine the approaches of compressed sensing and adaptive intelligence and evaluate them on the cross-anatomy inference task.

## 2. Methods

We propose to compensate for the artefacts induced by undersampled and/or downscaled $k$-space by Pix2Pix (Isola et al., 2018) and/or SRGAN (Ledig et al., 2017) models, the combination of which with a reconstructing U-Net allows for a fine control of the MRI acceleration factor (Belov et al., 2021). We trained these models on T2 slices of the fastMRI dataset (Zbontar et al., 2019) (73,478 brain and 167,530 knee slices). The brain slices underwent min/max intensity normalization to their 2nd and 98th percentiles. Because the synovial liquid shifts the 98th percentile to anatomically relevant features on T2 images the

knee slices were min/max normalized using a window centered on the soft tissue histogram peak. The width of the window was computed using the distance between low intensity soft tissue pixel and the soft tissue histogram peak. Both datasets were splits into train / validation / test subsets as 60% / 20% / 20%, without spreading patients between subsets.

## 3. Results and discussion

Upon cross-testing our models on the anatomy on which they were *not* trained, we observed reduction of the reconstruction quality (Figs. 1 and 2). Noticeable are both the sharp drop of the Pix2Pix models (likely caused by the simplicity of its generator) and the smaller drops of the SRGAN models.

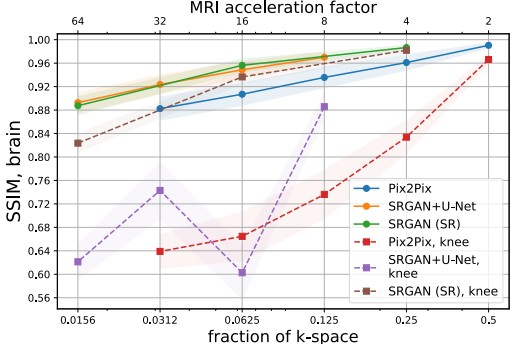

Figure 1: Models trained on brain slices.

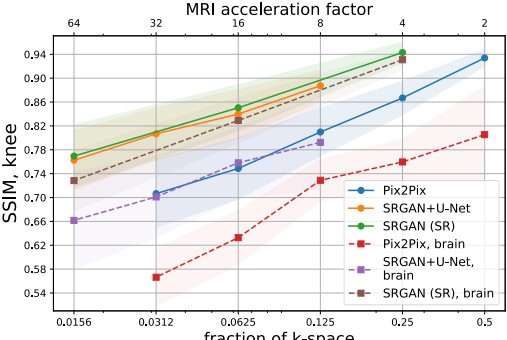

Figure 2: Models trained on knee slices.

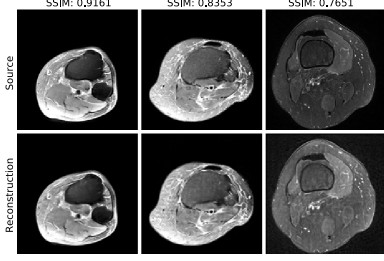

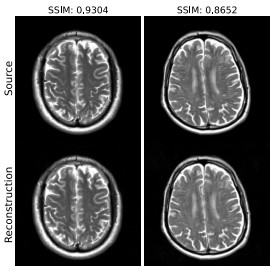

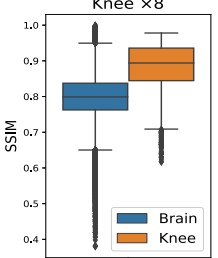

Figure 3: Knee reconstructions by SRGAN+U-Net trained on the brain data at ×16 acceleration.

Figure 4: Brain reconstructions by SRGAN+U-Net trained on the knee data at ×8 acceleration.

Figure 5: Distribution of image quality of SRGAN+U-Net trained on the knee at ×8.

The SRGAN+U-Net combo, when trained on the brain dataset and tested on the knee dataset, yields notably worse results at ×16 (or 6.25% of the $k$-space) than at other ac-

celeration factors (Fig. 1). Visually, however, except for a change of the brightness, the reconstructions still maintain satisfactory appearance (Fig. 3), which could be acceptable, *e.g.*, for such applications as radiation therapy planning. The brightness mismatch in training and inference leads to the big drops in SSIM but does not affect the perception of the image reconstructions. We, thus, stress the impact of the proper normalization on the cross-anatomy inference task.

Models trained at more moderate acceleration factors exhibit similar behaviour, *e.g.*, the SRGAN+U-Net preserves most of the anatomical structure present in the source image (Fig. 4). However, as distributions in Fig. 5 show, the majority of the inferior reconstructions have had noticeably brighter white matter on the slices (see, *e.g.*, Fig. 4, right).

We conclude that the SRGAN+U-Net combo can accelerate both the brain and the knee MRI scans. Importantly, its cross-anatomy generalization demands proper intensity normalization. In the future work, we foresee enhancement of our superresolution architecture by the maximization of the similarity between the histograms of different anatomic sites, *e.g.*, via the optimal transport theory or other statistical approaches.

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
