# OpenReview forum: "Cross-anatomy Inference of Deep Learning-based MRI Acceleration Methods"
_MIDL.io/2021/Conference/Short — Submitted to MIDL 2021_

### Official Review · Reviewer_TALC · 2021-04-20

**Confidence:** 4
**Final Rating:** 1

**Summary:**

This short paper studies the problem of cross anatomy inference of DL models for MRI reconstruction. A model was used to reconstruct MRIs of anatomic parts which are different from the anatomy used when training the recon model.  The authors claim that to compensate for the artefacts induced by undersampled and/or downscaled k-space, Pix2Pix and/or SRGAN models can be combined with a reconstructing U-Net and achieve a finer control of the MRI acceleration factor

**Strengths:**

This study uses different anatomies available in the fast MRI dataset. The study is interesting because it is somewhat suspected that reconstruction in the image to image setting, tend not to generalize to different anatomies, arguably because the network are not learning the image reconstruction physics.

**Weaknesses:**

The nature of short papers does not give room for extensive details, true, but I believe that some details which are crucial to appreciate the work are missing, and since the Authors do not provide access to their code, understanding is hampered because this makes it impossible to supplement the text by extracting details from the code.
It is in my opinion difficult to understand if the model acts in the image or in the k-space domain. Furthermore, it is not clear whether a physics-based approach is used, if consistency terms are included, and more importantly if models are learning the reconstruction physics. Figure 1 and 2 are misleading or maybe inverted because they show performance on the same training anatomy, as opposed to what is stated in the main text.

**Deanonymize Review:**

no

**Detailed Comments:**

Typically, the reconstruction on different anatomies degrades. This is especially true when the reconstruction is conducted in the image space. However, when the k-space information is exploited, this is no more valid, because the inverse problem is learned using the physic behind MRI reconstruction, and there exists a rich literature showing cross-anatomy reconstruction including papers from Johnson et al. at Miccai 2019, and more recently Yaman et al. 2021 appeared in Arxiv.

**Justification Of The Rating:**

For short papers, I strongly support new studies, even at their very early stage, but I believe this is not really providing any new knowledge to the field neither opens to new research directions. As reported  above, there exists already literature showing cross-anatomy reconstruction and conclusions in those works are more convincing.

**Paper Type:**

validation/application paper

**Special Issue:**

no

---

### Official Review · Reviewer_zaro · 2021-04-28

**Confidence:** 4
**Final Rating:** 3

**Summary:**

The work is about cross-anatomy inference of deep learning for MRI super-resolution. The network trained on anatomy-specific data faces performance drop when tested on another anatomy data due to domain shift. It found that proper intensity normalization could improve the performance of cross-anatomy inference.

**Strengths:**

The paper did extensive experiments using different network architectures (pix2pix, SRGAN, SRGAN+U-Net) and different acceleration rates to demonstrate the performance drop when training and testing data from different anatomies (brain, knee). These experiments is very helpful to know the domain generalization / domain adaption issue in deep learning for medical imaging.

**Weaknesses:**

1. The discussion part focused on SRGAN+U-Net. From Fig 1&2, SRGAN has the smallest performance drops in the cross-testing. It would be better to discuss more about SRGAN.
2. The author emphasized the impact of the proper normalization on the cross-anatomy inference task using Fig 3. It could be better to have a figure of displaying the intensity histogram of training data and testing data.

**Deanonymize Review:**

no

**Detailed Comments:**

1. From Fig1&2, SRGAN has smaller drops in cross-testing. Could the author give more discussion this part?
2. The authors compared Pix2Pix, SRGAN, and SRGAN+U-Net. I would like to some experiments to disentangle the network architecture and network loss, so we can better know the which part plays more in the  performance drop in cross-anatomy inference.
3. The authors said proper intensity normalization is important. Could the author give failure examples of improper intensity normalization, which might be helpful for readers better understanding this part?
4. The authors could explore more advanced domain adaption techniques in the future.

**Justification Of The Rating:**

The papers conducted lots of experiments to show the cross-anatomy inference / domain shift problem in MRI. Three networks are compared and each showed different performance drop in the cross-testing. However, the authors don't discuss the reasons. In addition, the authors said that proper intensity normalization is important for cross-testing but failed to give more details about this part.

**Paper Type:**

methodological development

**Special Issue:**

no

---

### Meta-Review · Area_Chair_WR4V · 2021-05-07

**Recommendation:** Reject
**Confidence:** 5

**Metareview:**

The AC does not follow all of the criticism; however, the AC agrees that the paper leaves open some important questions about the methodology and about the particular claimed novelty.

---

### Decision · Program_Chairs · 2021-05-11

Reject